# Climatic niche conservatism in non-native plants depends on introduction history and biogeographic context

Anna Rönnfeldt [1] ✉, Valén Holle [1], Katrin Schifferle [1], Laure Gallien [2], Tiffany Knight [3,4,5,6], Patrick Weigelt [7,8], Dylan Craven [9,10], Juliano Sarmento Cabral [11,12] & Damaris Zurell [1]

Niche conservatism is a fundamental assumption in predictive models for managing non-native species, but its generality remains debated due to mixed empirical evidence. We argue that this reflects underexplored context dependencies, as few studies have compared the niche dynamics of species introduced to multiple regions. Here, we quantify climatic niche changes in 1566 introductions of 316 non-native plant species across eight regions, including continents and archipelagos. While niche expansion into previously unoccupied conditions was low, niche conservatism and unfilling varied strongly across regions. Species with small native range sizes exhibited greater niche expansion. Longer residence times reduced niche unfilling, suggesting that a lack of niche conservatism observed in many regions might be transient and potentially linked to dispersal limitations. Our results highlight the necessity to consider region-specific contexts when assessing the potential for niche changes and provide a critical foundation for improving predictive models informing the management of non-native species.

Model-based risk assessments, crucial for the proactive management of biological invasions, often rely on the assumption that species occupy the same climatic space in their native and non-native ranges: niche conservatism[1]. Thus, our ability to predict the potential establishment of non-native species is entangled with the question of whether species conserve their niche when introduced elsewhere. To date, empirical evidence for or against niche conservatism in non-native ranges is controversial[2–6], but methodological differences across studies[7] and taxonomic differences make direct comparisons difficult.

We argue that context dependency may be the principal reason for the mixed results. This remains under-explored, as few studies compare the climatic niche changes of species introduced to multiple regions[8].

Species introduced to new ranges are considered niche-conservative when they occupy the same conditions in both the native and non-native ranges and thus show high niche stability (Fig. 1)[9]. Incomplete colonization or dispersal barriers that hinder species from occupying suitable conditions can result in a smaller subset of their climate niche in their introduced range, known as niche

[1]Institute for Biochemistry and Biology, University of Potsdam, Potsdam, Germany. [2]Laboratoire d'Ecologie Alpine, University Grenoble Alpes, University Savoie Mont Blanc, CNRS, LECA, Grenoble, France. [3]Department of Species Interaction Ecology, Helmholtz Centre for Environmental Research—UFZ, Leipzig, Germany. [4]German Centre for Integrative Biodiversity Research (iDiv), Halle-Jena-Leipzig, Leipzig, Germany. [5]Institute of Biology, Martin Luther University Halle-Wittenberg, Halle (Saale), Germany. [6]Department of Science and Conservation, National Tropical Botanical Garden, Kalāheo, HI, USA. [7]Department of Environmental Science, Radboud Institute for Biological and Environmental Sciences (RIBES), Radboud University, Heyendaalseweg 135, Nijmegen, The Netherlands. [8]Biodiversity, Macroecology & Biogeography, University of Göttingen, Göttingen, Germany. [9]GEMA Center for Genomics, Ecology & Environment, Universidad Mayor, Santiago, Chile. [10]Data Observatory Foundation, ANID Technology Center No. DO210001, Santiago, Chile. [11]School of Biosciences, College of Life and Environmental Sciences, University of Birmingham, Birmingham, UK. [12]Ecological Modelling, Bonner Institute for Organismal Biology—Department of Plant Biodiversity, University of Bonn, Bonn, Germany. ✉e-mail: anna.roennfeldt@uni-potsdam.de

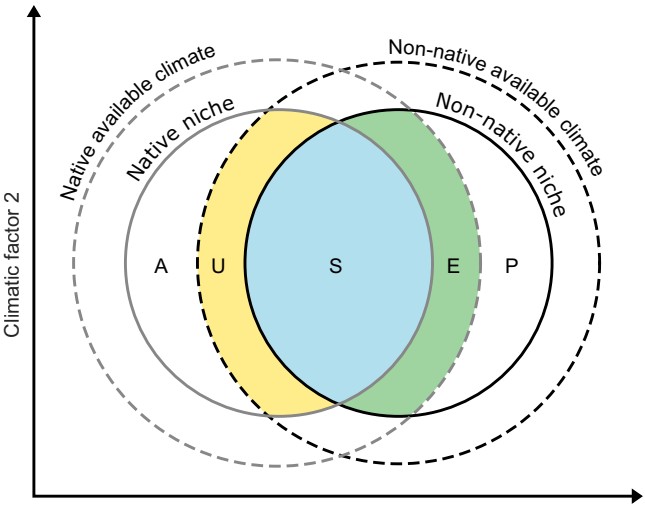

**Fig. 1 | Conceptual overview of niche dynamics between the native and non-native ranges, and corresponding niche change metrics.** The solid lines indicate the native (gray) and non-native (black) climatic niches and the dashed lines the available climatic conditions in the native and non-native range. In analog climatic conditions, niche stability (S, blue) describes the parts of the climatic niche occupied in both ranges, unfilling (U, yellow) indicates native niche space that is not part of the non-native niche space, and expansion (E, green) refers to non-native niche space that is not part of the native niche space. Under non-analog conditions, abandonment (A) indicates environmental conditions within the native niche that are not available in the non-native range, and pioneering (P) describes the part of the non-native niche that has no analog in the native range. Figure adapted from Fig. 1 in Guisan et al.[9].

unfilling[5,7]. Expansion into previously unoccupied parts of the climatic niche space occurs, e.g., due to changes in biotic interactions[10,11], the absence of dispersal barriers present in the native range, or rapid adaptation occurring after the introduction to a new region[12]. To understand the context dependency of niche dynamics, niche change metrics are quantified in analog niche space (i.e., where similar conditions are present in both ranges), as metrics related to non-analog conditions (abandonment and pioneering, Fig. 1) are difficult to compare between species and regions.

We hypothesize that multiple factors could lead to regional differences in niche dynamics (Table 1). Higher dispersal ability, reflected by traits associated with long-distance dispersal, could aid non-native range expansion[13] and result in relatively lower niche unfilling and, thus, higher niche stability, compared to dispersal-limited species. Also, dispersal limitations could be mediated and niche unfilling reduced by longer residence times, as was shown in non-native vertebrate species[14]. Additionally, species with small native ranges and narrow climatic niches could show higher niche expansion as they are released from non-climatic niche and range constraints, such as biotic limitations[15]. While ecological traits are intrinsic to species, their biogeographic attributes and introduction history may vary across non-native regions, increasing the likelihood of regional differences in niche dynamics. Species introductions to islands, for example, could result in more pronounced differences in the niche dynamics compared to mainland regions, as their geographic isolation could lead to dispersal limitation and higher niche unfilling. Moreover, depauperate species communities of islands could result in higher niche expansion due to biotic constraints potentially being lifted[16]. One such island system is the Pacific Island region, which harbors the highest number of invasive plant species per unit area[17]. Many of these species have also been introduced to other regions[18], which makes for an ideal natural laboratory to compare niche dynamics of non-native plants across multiple regions.

**Table 1 | Overview of the hypothesized factors that might affect niche unfilling (U), stability (S) or expansion (E) of non-native plants upon their introduction to a new range**

| Factor | Predictors | Hypothesized effect | | | Hypotheses |
|---|---|---|---|---|---|
| | | U | S | E | |
| Ecological traits | Plant height, seed mass, and growth form (herb<shrub<tree) as dispersal proxies[59,61] | - | + | | Species capable of longer dispersal distances are better at overcoming dispersal barriers and thus show lower niche unfilling and higher niche stability sensu[13]. |
| | Life cycle length | - | + | - | Species with a shorter life cycle show higher niche expansion due to higher potential for rapid adaptation and lower niche unfilling as new generations represent opportunities for further dispersal throughout the non-native range. |
| Introduction history | Time since introduction | - | + | | Species with longer residence times show lower niche unfilling and higher niche stability because they had more time to disperse throughout the non-native range[14]. |
| Biogeographic attributes | Climatic niche breadth of native niche | + | | - | Species with a narrow climatic niche show higher niche expansion due to a release from non-climatic restraints[33]. |
| | Range size of native range | + | | - | Species with a smaller native range show higher niche expansion due to a release from non-climatic restraints[33]. |
| | Latitudinal distance between range centroids | - | | + | Longer latitudinal distances between native and non-native range centroids lead to lower niche stability due to differences in biomes and community assemblages along the latitudinal diversity gradient[72]. |
| | Native climatic niche characteristics (expressed as the centroid along PCA axis 1, representing a warm to cold gradient, and along PCA axis 2, a wet to dry gradient) | | +/- | | Niche stability varies based on how the native climate compares to the conditions in the non-native region (including local community assemblages and other non-climatic limitations in the non-native region sensu[30,31]. |

The hypothesized effects of the predictors on each niche change metric can be positive (+), negative (-), mixed (+/-) or non-existent.

We work with 316 plant species from the PaciFLora data set[19] that have been introduced to at least two of the eight study regions globally, including the Pacific Island region, compile georeferenced occurrence data across regions and match them with a biogeographic status (native, non-native). We then use an ordination-based approach to quantify region-specific climatic niche dynamics, and phylogenetic multiple regression to assess the relationship between niche change metrics and ecological and biogeographic attributes, as well as residence time.

In this study, we show that there are distinct regional differences in the niche dynamics of non-native plant species introduced to multiple regions, and that these depend on their region-specific residence times and the overall biogeographic context. Species with smaller native range sizes exhibit greater niche expansion, while niche unfilling decreases, and niche stability increases with increasing residence time. The latter indicates that the lack of niche conservatism observed in many cases is likely transient.

## Results

### Niche conservatism

We followed the methodology of Broennimann et al.[20] to quantify native and non-native niche spaces along the first two axes of a principal component analysis (PCA) including 19 bioclimatic variables, in order to assess niche conservatism and to quantify niche change metrics between pairs of native and non-native niches. We focused our analysis on plant species that have been introduced to the Pacific Islands, and at least one of seven other botanical continents[21]: Africa, Australasia, Europe, North America, South America, temperate Asia, and tropical Asia. In total, we compared 1566 regional introductions for 316 non-native plant species. The native ranges of these species covered several climatic zones (classified based on the main Köppen climate zones)[22] and regions (SI appendix). Most commonly, species' native ranges were found in tropical to temperate climate zones, and occur in equal proportions in the considered non-native regions, except that fewer temperate species were introduced to tropical Asia and fewer tropical species were introduced to Europe (Fig. S2).

Using an ordination-based approach, we compared the climatic niches found in the native range and in each non-native region and tested for niche conservatism and niche switching using niche similarity tests. We observed significant niche conservatism for 45.5% of the introductions and found no indication of niche switching (Fig. S3). For the remaining regional introductions, the similarity tests were not statistically significant, meaning that these species neither conserve nor switch their climatic niche. Notably, most species showed inconsistent patterns of niche conservatism across the regions they were introduced to, with significant niche conservatism in some but not all non-native regions (Fig. 2A). Only 20 species consistently conserved their climatic niche across the regions they were introduced to, while 47 showed no niche conservatism in any region (Fig. 2A, Supplementary Data 1). The lowest proportion of species with significant niche conservatism within an individual region was found on the Pacific Islands, with only 25% of the non-native species conserving their climatic niches after their introduction to the region (Fig. 2B).

### Niche change metrics

For each species and region, we compared the native and non-native niche and quantified the following niche metrics in analog climate space: niche stability, niche expansion and niche unfilling (cf. Fig. 1, Supplementary Data 1). Niche stability was high across species and regions, with species' niches in native and non-native range overlapping by a median of 78.4% across all introductions (Fig. 3), characterizing the biggest share of the analog niche space. Regarding niche space only occupied in one region, niche unfilling was considerably higher (median 17.6% across all introductions) than niche expansion

(median 0.78% across all introductions). As for regional differences, ANOVA results indicated significantly higher niche unfilling for plants introduced to the Pacific Islands compared to all other regions ($F_{(7,1558)} = [27.64]$, $p < 0.001$), except North America (Fig. 3, Table. S1). Consequently, niche stability in species introduced to the Pacific Islands was often lower compared to the other regions. Niche expansion was highest in tropical Asia (median 3.4%) and some outliers were observed in each region, with some species showing niche expansion exceeding 50%. Additional results for niche abandonment and pioneering in non-analog climate space are provided in the SI appendix (Figs. S4, S10, S11, S12) and Supplementary Data 1.

### Trait analyses

We used multiple phylogenetic regression to assess the effect of ecological traits, biogeographic attributes and residence time on region-specific niche change metrics while controlling for shared evolutionary history between species. Due to data gaps, these analyses were restricted to 165 of the 316 plant species for which information on all traits was available, covering 770 of the 1566 regional introductions in our study. Species from different climate zones had similar mean trait values, except for a wider mean niche breadth in temperate species and a higher mean seed mass in tropical species (Fig. S9). Separate models were fitted for each niche change metric and region. Trait models explained 7–52% of the variance in niche unfilling (Fig. 4A), 17–56% in niche stability (Fig. 4B), and 17–55% in niche expansion (Fig. 4C). The relative importance of different traits for driving niche change metrics varied considerably across regions. Biogeographic attributes and species residence time were overall more important than ecological traits in explaining niche changes (Fig. 4A–C): the combined variable importance of ecological traits only accounted for a median of 26.75% for unfilling, 29.0% for stability, and 31.9% for expansion. A few general patterns emerged: niche unfilling consistently decreased and niche stability increased for species with longer residence times (Fig. 4A, B). Also, niche expansion consistently decreased for species with larger native range sizes and increased with longer distances between latitudinal range centroids (Fig. 4C). These patterns also emerged in the univariate models, indicating that they are robust to missing species in the multi-trait analyses (Fig. S12).

Among the biogeographic attributes, niche centroids were generally most important for explaining niche unfilling and niche stability, but the direction of these effects was variable (Fig. 4). For example, species native to colder climates (niche centroid 1) showed higher niche stability and less niche unfilling when introduced to temperate Asia and North America, but lower niche stability and greater unfilling when introduced to tropical Asia and the Pacific Islands. Species native to dry climatic conditions (niche centroid 2) showed lower niche stability and more niche unfilling when introduced to Europe and the Pacific Islands, and higher niche stability and lower niche unfilling when introduced to tropical Asia. While native range size negatively affected niche expansion in all regions (Figs. 4D–F, S12), it was rarely selected in the final trait models for niche stability and niche unfilling (Fig. 4). Also, the effects of the native climatic niche breadth and latitudinal distance between native and non-native regions were minor. If selected in the final models, however, climatic niche breadth had a negative effect on niche stability and a positive effect on niche expansion and niche unfilling (Fig. 4A–C). For the ecological traits, no generalities emerged. Where life cycle was included in the final models, niche unfilling tended to decrease and niche stability to increase for longer-lived species, except in temperate Asia, where the opposite was observed. Niche expansion, surprisingly, increased with plant height in a few regions (Fig. 4C). The phylogenetic signal of niche change metrics was close to zero in most cases, indicating that they were not conserved phylogenetically, except for a few notable regional exceptions (Table S2).

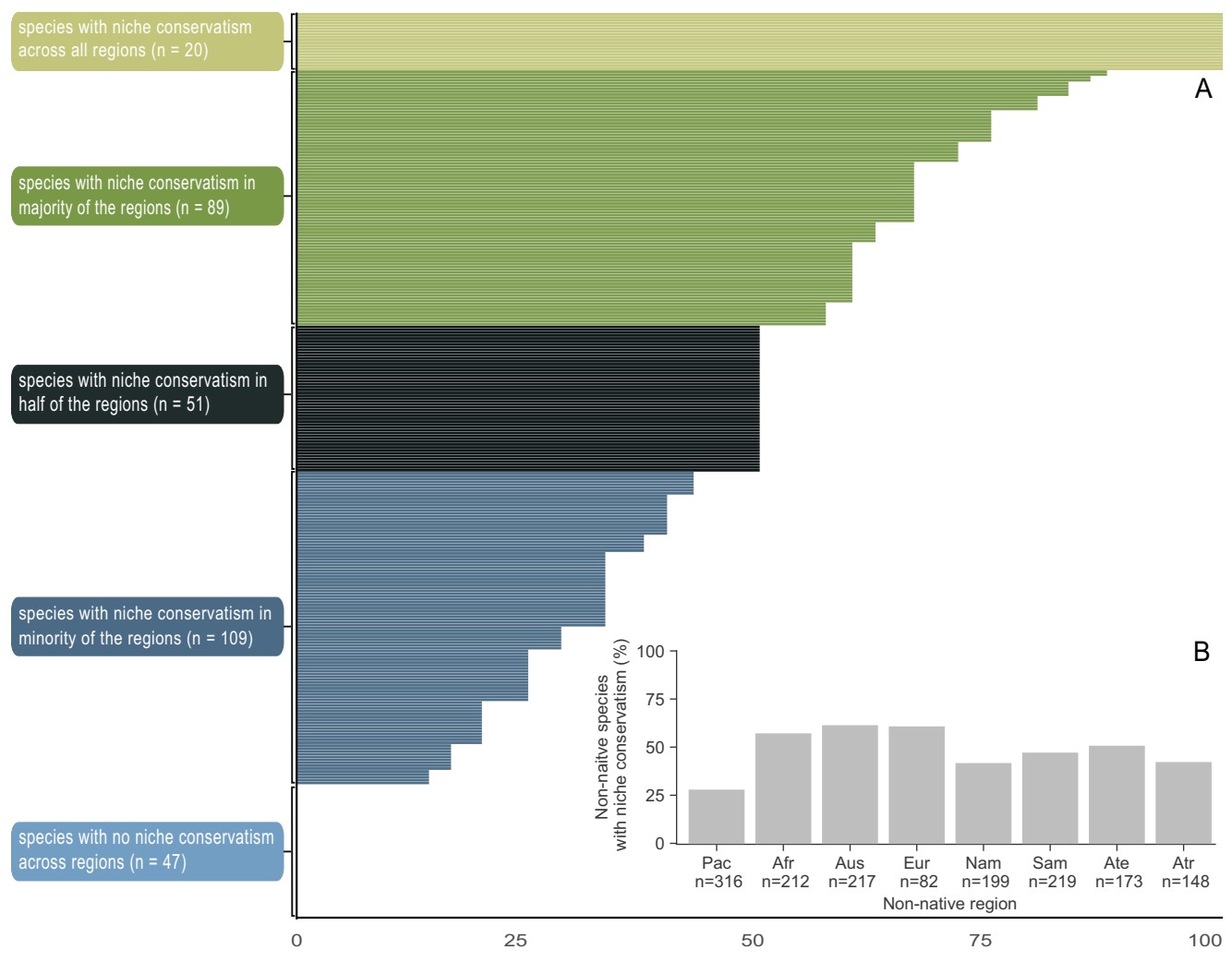

**Fig. 2 | Niche conservatism across regions.** Similarity tests were used to test for significant niche conservatism in non-native vascular plants introduced to different regions. In **A** each horizontal line represents one species and the percentage of regions in which that species showed significant niche conservatism based on null models ($n = 1200$ iterations). Colors indicate whether species conserved their niche in all (light green), the majority (>50%, but <100%, dark green), half (black), the minority (<50%, but >0% dark blue) or none of the regions in which they have been introduced. **B** shows the percentage of species that conserved their niche for each non-native region, respectively (Pac = Pacific Islands, Afr = Africa, Aus = Australasia, Eur = Europe, Nam = North America, Sam = South America, Ate = temperate Asia, Atr = tropical Asia). Note that no species showed statistically significant niche switching (Fig. S3). Source data are provided as a Source Data file.

## Discussion

Our analysis quantifying the climatic niche dynamics of non-native plants introduced to different world regions revealed that climatic niche conservatism is highly context-dependent and largely depends on residence time. Niche change metrics strongly varied across regions. Niche stability was generally high, although substantial niche unfilling occurred while niche expansion was consistently low. Using null models, we found that the same species may show significant niche conservatism in some regions but not in others. We further found that biogeographic attributes and residence time affected niche change metrics more strongly than ecological traits linked to dispersal and life cycle. This suggests that climatic niche conservatism depends, at least to some extent, on the climatic preferences of introduced species and that deviations from climatic niche conservatism may be transient.

We found strong variation in climatic niche change metrics across regions, indicating that regional inconsistencies in niche dynamics are common among introduced plants. This corroborates previous findings from multi-region niche comparisons of a limited number of plant species[8,23]. Niche unfilling was prevalent across all regions, which could

be due to geographic barriers hindering dispersal or biotic constraints. It could also indicate transient dynamics resulting from ongoing dispersal processes after the species' initial introduction[5,7]. As hypothesized, we found that niche unfilling was higher for introductions to the Pacific than other regions, likely due to the barrier effects of the ocean and the geographic isolation between archipelagos. It suggests that many of the species introduced to the Pacific Islands have not yet reached their climatic equilibrium and have unrealized colonization potential[24]. It is thus likely that human-mediated transport, which often facilitates inter-island dispersal in the Pacific region[18], will eventually enable some of these species to fill their potential range, further increasing biological invasions across this region.

Niche expansion was generally low, indicating that competitive release or niche adaptation is not prevalent across species and regions. In the case of the Pacific Islands, where we expected to see more niche expansion because of the depauperate species communities of islands[16], the geographic fragmentation of the region could have further contributed to the species not being able to reach and thus expand into otherwise available niche spaces. To understand the specific mechanisms behind the low expansion values, a smaller-scale

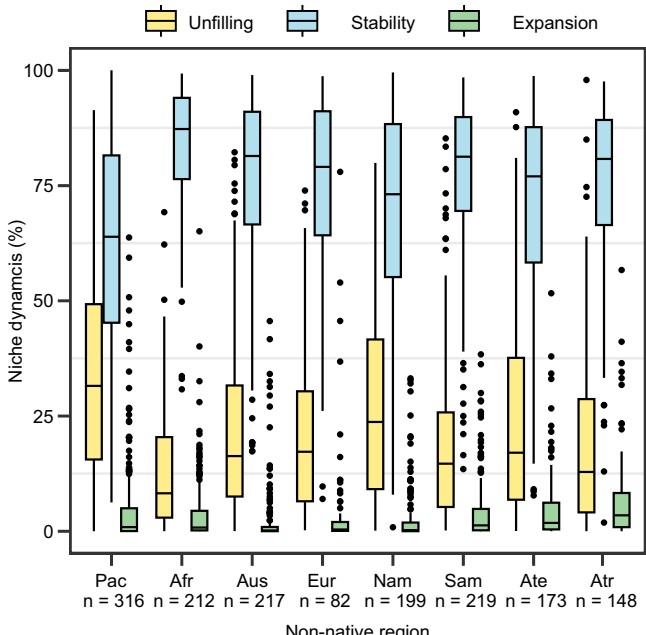

**Fig. 3 | Climatic niche dynamics in non-native plants across regions.** The box outlines indicate the interquartile range (IQR), and the horizontal lines represent the median values. The whiskers extend no more than 1.5 times the IQR from the boxes, with individual points identified as outliers. The sample size in each region indicates the number of species that have been introduced to each region. Source data are provided as a Source Data file.

study would be more suitable, focusing on factors such as regional community assemblages, total climatic expansion potential[25], or anthropogenic transport networks facilitating inter-island dispersal[18]. When comparing the overall low niche expansion with previous studies that quantified niche differences across several continents, our results stand in stark contrast to results by Atwater et al.[2] who found niche expansion to be common in introduced plants. Our results align with those of Petitpierre et al.[5] who also found lower niche expansion compared to niche unfilling. Low niche expansion would be a promising sign for the reliability of model-based risk assessments that typically rely on the assumption that the realized niche is adequately captured by available data[1,26], especially for non-native species where no georeferenced data for previous introductions in other regions are available. However, it is important to note that relatively low niche expansion in climatic space could still result in a high species prevalence in geographic space if those climates are common[27]. This could lead to an underestimation of the establishment potential of non-native species[28] and thus to uncertainty in management planning and decision-making. While overestimations rather than underestimations are more desirable for risk assessments, pronounced niche unfilling in geographic space can likewise affect management decisions if the costs of preventive measures across larger areas would drastically outweigh the eventual negative impacts of a species should it fill its niche in these areas[29].

Despite consistent evidence of climatic niche unfilling and, to a lesser degree, niche expansion across species and regions, we did not detect niche switching. Still, we confirmed statistically significant niche conservatism in less than half of the considered introductions, and we found many cases where species showed statistically significant niche conservatism in one or few, but not all, regions. This implies that the contradictory results found in the existing literature on niche conservatism for introduced species might not only be a result of methodological or species-specific differences[7,9] but also due to biogeographic differences across focal regions. Although recent

reviews indicate that niche conservatism is the norm when species are introduced to a new region[7], we did not find direct support for this, as we found that niche conservatism was not a general trend for plant species introduced to multiple regions. We do, however, suggest that a lack of niche conservatism is likely transient for many of these introductions, as our trait analysis revealed a strong connection between niche unfilling and a species' residence time in a particular region. This corroborates previous findings by Strubbe et al.[14] and emphasizes the transient nature of niche unfilling. Interestingly, although residence time also had a negative effect on niche unfilling for introductions to the Pacific Islands, it was not selected in the final models, indicating that compared to other regions residence time was less important in the Pacific. This is likely due to the geographic isolation between islands, which drastically limits propagule dispersal within the region. For example, *Ricinus communis* (L.), one of the species with the longest residence times in our data set, only showed 1.6% of niche unfilling in North America, 201 years after its initial introduction. Only 3 years prior, *R. communis* was first recorded on one of the Pacific Islands, yet, unfilling still contributes 43.4% to the analog niche space of the region. Overall, the high prevalence of niche unfilling and associated dispersal limitations and the strong role of residence time for explaining these niche dynamics further corroborates that niche conservatism is a matter of time and that most non-native species could eventually tend towards climatic niche conservatism. Likely exceptions to this are introductions to archipelagos, unless anthropogenic activities can relax inter-island dispersal limitations.

Our trait analyses confirmed that niche conservatism is highly context-dependent. Specifically, native climatic niche position had a consistently strong effect on niche stability and niche unfilling. Thus, niche stability is to some degree determined by the overlap between a species' climatic preference and the main climatic conditions found in non-native regions. Here, we focused on the analog niche space, so this does not refer to the presence or absence of particular climatic conditions. Rather, this connection likely reflects the importance of factors such as the occurrence of analog climatic conditions in the geographic space (common vs. marginal) and resulting non-climatic limitations (e.g., biotic interactions[30,31] or land use intensity[32]) for the establishment and thus niche stability of introduced species. This context dependency also emphasizes that observed niche changes in one region cannot be easily generalized to other regions. Another important biogeographic attribute affecting niche change was native range size, which was often the most important variable per region. Species with smaller native range sizes, such as *Opuntia ficus-indica* (L.) native to Mesoamerica, showed comparably high niche expansion values in all eight regions. This result coincides with those of other studies that have found a negative correlation between native range size and niche expansion[33,34]. This indicates that the native ranges of these species were not constrained by climatic conditions[35], and that it is more likely that it was the release from non-climatic constraints (biotic or geographic) that have enabled these species to expand their climatic niche in the non-native regions[10]. The importance of biotic interactions for niche expansion will likely become even more relevant in the future, as climate change can affect the competitive abilities of both native and non-native species[36]. This could potentially result in regional niche switching if a non-native species is at an extreme competitive advantage compared to native species.

The effects of other biogeographic traits on niche expansion and niche unfilling were less pronounced and varied more strongly across regions. Similar to range size, native niche breadth has previously been found to be negatively correlated with niche expansion[33]. In our analyses, however, niche breadth was only relevant for explaining niche expansion in two regions, and the direction of the effect was contrary to previous reports[33], with niche expansion increasing with climatic niche breadth. Also, we found little effect of ecological traits on the niche change metrics. We expected to observe less niche unfilling for

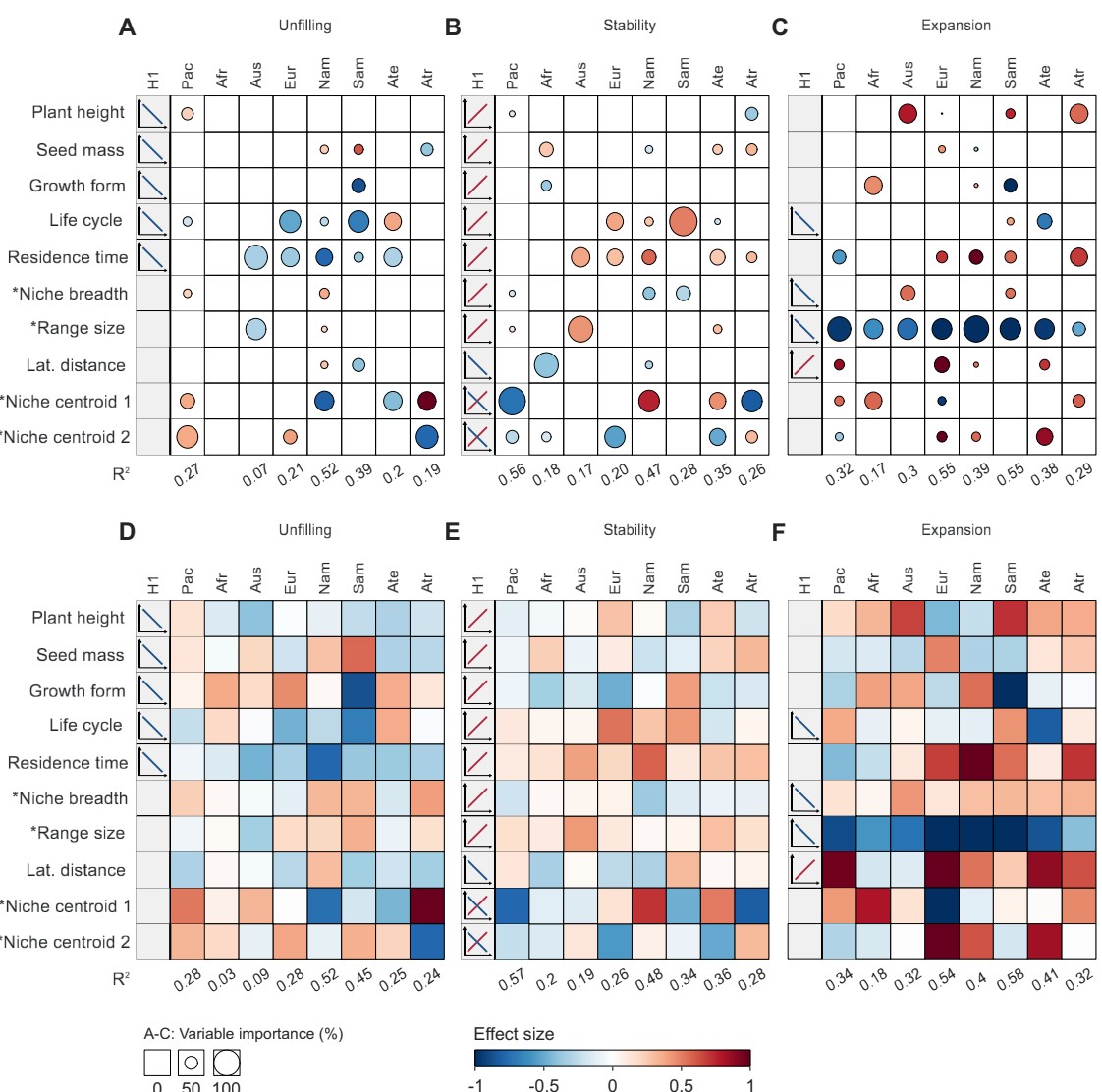

**Fig. 4 | Trait effects on climatic niche changes in non-native plants.** For each non-native region, we fitted regression models with AIC-based stepwise selection for **A** niche unfilling, **B** niche stability and **C** niche expansion. The first column (H1) in each panel shows our hypotheses about the expected relationship (see Table 1). All other columns represent the final model for each region, with the total explained variance ($R^2$) shown below the columns. Circle sizes indicate the variable importance (%) of single traits within the multiple regression models, and the effect size shows whether the respective niche change metric increases (red) or decreases (blue) as the trait values increase. **D**–**F** show the corresponding full models. Traits labeled with an asterisk refer to biogeographic traits estimated for the native range or native niche of the species. Niche centroids refer to the relative position along climatic gradients 1 (from warm to cold) and 2 (from wet to dry). The sample size varied between regions: Africa (Afr, $n = 124$), temperate Asia (Ate, $n = 95$), tropical Asia (Atr, $n = 78$), Australasia (Aus, $n = 124$), Europe (Eur, $n = 56$), North America (Nam, $n = 110$), Pacific Islands (Pac, $n = 143$), South America (Sam, $n = 41$). Source data are provided as a Source Data file.

species capable of longer dispersal distances and greater niche expansion for species with shorter life cycles, but neither of these hypotheses was unequivocally corroborated by our results. This aligns with previous research by Early and Sax[33], who also found little effect of dispersal capacity on niche unfilling. Unexpectedly, life cycle played an important role for niche unfilling and stability in some regions, while plant height was included in some of the final niche expansion models. However, in both cases, no general patterns emerged across regions regarding their effect on niche change metrics. Overall, our trait analysis revealed that there are only a few general factors that explain climatic niche dynamics for non-native plants across regions. In particular, biogeographic attributes were the most important variables for explaining niche conservatism. Previous studies have shown that biogeographic attributes are also critical determinants of the establishment success of introduced species[37,38].

Multi-region comparisons of niche dynamics in non-native species are a major challenge, and our analyses relied on standardizing heterogenous data sources for native and non-native occurrence data, and trait data. Given limitations of trait data availability[39], we selected ecological traits that aligned with our hypotheses while maximizing species coverage in the analyses. While the phylogenetic signal in the regional models was generally low, the stronger signals observed in a few cases suggest that additional ecological traits, beyond those considered here, may contribute to regional differences in niche dynamics. Future studies could benefit from incorporating a broader range of ecological traits, including those related to competitive ability (such as clonality and allelopathic potential) and dispersal potential. Ideally, the prevalence of these traits within the native communities should also be assessed to provide insights into potential niche opportunities or biotic resistance[40], particularly at finer spatial scales. Working at a

subcontinental scale could further allow the inclusion of more detailed data on anthropogenic disturbances, land use, or horticultural history. Such information could help explain unexpected trait effects observed in some regions and provide further insights in the drivers of regional niche differences among non-native plants.

Here, we provide evidence that climatic niche conservatism may be a common endpoint of plant introductions into non-native regions, and many species show greater niche unfilling, which is indicative of dispersal limitations rather than niche expansion. This lends strong empirical support for model-based risk assessments for the preventive management of biological invasions. Nevertheless, some species can reach high levels of climatic niche expansion that appear to be related to changes in the realized climatic niche. Still, local adaptations, particularly on islands, cannot be completely discarded. Importantly, our results suggest that earlier contradictory findings of niche conservatism vs. niche switching are largely related to context dependency and introduction history. Future research should focus on studying the temporal aspects of climatic disequilibria in non-native species.

## Methods

We used R version 4.2.2[41] for all analyses and visualization. A full list of all R packages used is provided in the SI appendix and in the repositories that store the code needed to reproduce our analyses.

### Study region

We focused on non-native plant introductions to eight distinct regions: the Pacific Islands, Africa, Australasia, Europe, North America, South America, temperate Asia, and tropical Asia (Fig. S1). Except for the Pacific Islands, these regions correspond to level 1 of the world geographic scheme of recording plant distributions[21]. The spatial delimitation of the Pacific Islands was based on Wohlwend et al.[19] covering 50 island groups located in the Pacific Ocean between 40°N and 40°S.

### Species occurrence data

The initial species list consisted of the 3962 vascular plant species listed by the PaciFLora data set[19]. PaciFLora builds on PIER (Pacific Island Ecosystems at Risk, http://hear.its.hawaii.edu/Pier/) and GloNAF (Global Naturalized Alien Flora)[42] and lists vascular plant species with known naturalized occurrences on at least one of the islands in the Pacific region. This species list presented a suitable starting point for our analysis, as it allowed us to identify species from various plant families and biogeographic origins that have been introduced to this island system, as well as to other regions. We downloaded all available occurrence data for these species from GBIF (Global Biodiversity Information Facility, https://www.gbif.org/)[43] and BIEN (Botanical Information and Ecology Network, https://bien.nceas.ucsb.edu/bien/)[44–49] in June 2023, using the R packages *rgbif*[50] and *BIEN*[51]. After merging these data at a 1 km resolution, we cleaned them by removing duplicates and occurrences with erroneous time stamps or coordinates, using the R package *CoordinateCleaner*[52]. We then sampled background data within a 200 km terrestrial spatial buffer around the presence points to represent biogeographically accessible areas, aiming for a ratio of ten times more background data points than presence points, if possible[53]. We performed a sensitivity check using a buffer size of 50 km for a random subset of 15% of the species ($n = 49$), as a smaller buffer size is potentially more realistic for island systems. Sampling background points within a smaller buffer reinforced our main findings (Figs. S5–S8), but we opted to use a 200 km buffer instead, as we consider it to be more plausible for the continental study regions. To reduce spatial autocorrelation, all presence and background data points were spatially thinned to a minimum distance of 3 km between points.

### Status assignment

Three data sources were incorporated to determine whether an occurrence point would be considered a native or non-native occurrence of the species in the respective regions. The first source was the WCVP (World Checklist of Vascular Plants)[54], accessed via the R package *rWCVP*[55]. WCVP provides status information at a spatial resolution corresponding to level 3 of the world geographical scheme for recording plant distributions[21]. To supplement the information provided by WCVP, we further included data from GIFT (Global Inventory of Floras and Traits)[56] and GloNAF[42]. As the three data sources used different labels, such as naturalized, alien, invasive, and non-native, we opted to only distinguish between *native* and *non-native* while harmonizing the information provided by the three sources. Conflicting status information was resolved (where possible) by using the status provided by the source that refers to a smaller spatial scale (e.g., a regional checklist rather than an entire country). Occurrences where conflicts between data sources could not be resolved or for which no matching status information was available were excluded.

### Climatic data and main climate zones

For the main analysis, we used all 19 bioclimatic variables from CHELSA V2[57,58] at a 1 km resolution. These variables describe means, extremes, and seasonality of temperature and precipitation. The CHELSA V2 data are based on a mechanistic terrain-based downscaling approach and thus facilitate improved regional climate space estimates. We further used the Köppen climate zones to classify the climatic zones in the species' native ranges[22]. A climate zone was considered to be among the main climate zones for a species if 30% of the species' native occurrences lie within that zone.

### Species trait data, residence time and biogeographic attributes

For subsequent trait analyses, we obtained data on ecological traits, biogeographic attributes, and the region-specific introduction history for each species. First, we considered four ecological traits: life cycle as a proxy for adaptive potential, plant height, growth form and seed mass as proxies for dispersal ability[59]. We opted to use these dispersal proxies, as other traits associated with dispersal, such as dispersal mode, were only available for a smaller subset of the species and could not be unambiguously translated into longer or shorter dispersal distances. We gathered data from the GIFT database, version 2.2, accessed via the R package *GIFT*[60]. Categorical trait values were transformed to an ordinal scale. Life cycle ranged from short to long life cycles: annual <biennial <perennial. Growth forms ranged from short to long-distance dispersal potential[59,61]: herb <shrub <tree. Mean plant height (m) and seed mass (g) were continuous.

Second, we quantified several biogeographic attributes for each species: native niche breadth and centroid, native range size, and the region-specific latitudinal distance between native and non-native range centroids. We quantified native niche breadth using the Shannon Index of the species' occurrence density in a two-dimensional climatic niche space[62,63] using global climate as background data to allow interspecies comparisons. Occurrence density was corrected for climatic availability using kernel density estimators. Also, we extracted native niche centroids along the first two PCA axes, using global climate data. The first axis describes a gradient from warm to cold conditions (niche centroid 1) and the second axis a wet-to-dry gradient (niche centroid 2). Native range size was defined as the total area of the WCVP level 3 polygons and/or GIFT polygons, with which native occurrence points were matched during the biogeographic status assignment. Latitudinal distance between native and non-native range centroids was calculated separately for each region to which species were introduced, and serves as a proxy for differences in biomes and community assemblages between native and non-native regions. Lastly, we extracted residence time from the *FirstRecords* database[64,65]. Residence time was

calculated as years since the first occurrence record in a region, with 2023 as the reference year. In cases where approximations instead of specific years were provided, we transformed these to numeric values: "early 20th century" to 1925, "mid-20th century" to 1950, and "late 20th century" to 1975.

## Niche analyses

For each species, we quantified climatic niches and assessed niche dynamics between each pair of native range and non-native region. We only considered regional introductions with at least 20 occurrence points within analog climatic conditions in the native and the non-native niches, respectively. A species was only included in the overall analysis if we could run the niche comparison between the species' native range and the Pacific Island region, as well as at least one other region. The final analysis covered 316 species and a total of 1566 regional introductions. Of the 316 species that have been introduced to the Pacific Islands, 212 were also introduced to Africa, 173 to temperate Asia, 148 to tropical Asia, 217 to Australasia, 82 to Europe, 199 to North America, and 219 to South America.

Niche dynamics were quantified using the R package *ecospat*[66], following the methodology of Broennimann et al.[20]. Our analysis consisted of three steps, performed for each introduction. First, the native and non-native realized climatic niches were determined by calculating the occurrence density, and the available environmental space (background data) along the first two axes of a PCA using all 19 bioclimatic variables using kernel density estimators. Next, we calculated niche overlap using Schoener's *D* metric[67] and performed similarity tests (with $n = 1200$ iterations). The similarity test was used to specifically test for niche conservatism (significantly higher niche overlap than expected by chance, $p < 0.05$) or niche switching (significantly lower overlap than expected by chance, $p < 0.05$) between the native and non-native niches. We tested these assumptions separately. Lastly, we calculated the niche change metrics in terms of stability, unfilling, and expansion. These were standardized to sum up to one, to enable us to assess the relative contribution of each metric to the total analog niche space. For completeness, we additionally quantified niche abandonment and pioneering in non-analog climates.

## Trait analyses

We assessed the effect of ecological traits, biogeographic attributes and residence time on region-specific niche change metrics using multiple phylogenetic regressions. Due to data gaps, only 165 out of the 316 species were included in these analyses, with different numbers of species considered in each region: Africa ($n = 124$), temp. Asia ($n = 95$), trop. Asia ($n = 78$), Australasia ($n = 124$), Europe ($n = 56$), N. America ($n = 110$), Pacific Islands ($n = 143$), S. America ($n = 41$). In total, the trait analyses considered 770 of the 1566 quantified niche pairs. All models were estimated using the R package *phylolm*[68], to account for potential biases due to phylogenetic relatedness between the species and their associated traits[69,70]. Phylogenetic data were extracted from the PaciFLora dataset and built on the seed plant phylogeny by Smith and Brown[71]. We fitted separate models for each niche change metric and region. In each case, the most parsimonious model was selected using stepwise, Akaike information criterion (AIC)-based variable selection. To determine variable importance, each variable was randomly permuted ($n = 99$) at a time, and the drop in explained deviance between the model with and without permutation was quantified. Variable importance across predictors was standardized to sum up to 100%. We additionally fitted univariate models with higher species sample sizes for each region to account for the robustness of the signals detected in the most parsimonious models against missing species. We also tested the inclusion of an interaction term between residence time and ecological traits in preliminary analyses. As the interaction terms did not significantly increase explained variance, we did not consider them in the final trait analyses.

## Reporting summary

Further information on research design is available in the Nature Portfolio Reporting Summary linked to this article.

## Data availability

The data generated in this study are provided in the Supplementary Information and Source Data file. The processed occurrence data, matched with biogeographic status information, are deposited on Zenodo (https://doi.org/10.5281/zenodo.16992420). The derived dataset of the initial GBIF download in 2023 can be accessed here: https://doi.org/10.15468/DD.685JDS. Publicly available datasets used in this study are: species list from PaciFLora (https://doi.org/10.3897/BDJ.9.e67318), occurrence data from GBIF and BIEN (accessed via R), status information from WCVP (accessed via R), GIFT (accessed via R) and GloNAF (https://glonaf.org/), trait data from GIFT (accessed via R), bioclimatic variables from CHELSA V2 (https://doi.org/10.5061/DRYAD.KD1D4), and the Alien Species First Records Database (https://doi.org/10.5281/ZENODO.10039630). Source data are provided with this paper.

## Code availability

The code required to reproduce this work is split across two public repositories. The first (https://doi.org/10.5281/zenodo.14850232) covers the download and cleaning of the occurrence data, including the biogeographic status assignment (native vs. non-native). The second (https://doi.org/10.5281/zenodo.16993239) covers the niche comparison, trait data collection, and trait analyses.

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

## Acknowledgements

We acknowledge funding support by the Deutsche Forschungsgemeinschaft (DFG, German Research Foundation), project 505288495 (AR, VH, DZ). We acknowledge the herbaria that contributed data to this work: U, MBML, UFRN, ARIZ, BRIT, HUAL, M, CSLA, CMC, CM, BRNU, BIRM, BHSC, BERN, BASBG, ABD, MAL, MACB, LU, LOB, LMU, LAU, KSP, KR, KATH, KAND, IBF, HSU, HLU, GSW, GJO, FSC, ERE, DES, DEE, TUCH, TOGO, TBI, TAA, SNUA, SJC, SHIN, RENO, PI, NMW, NMLU, NHG, NEBC, NCCE, MSUB, YRK, YA, WSCO, WCUH, WB, VPI, VALPL, UNSW, A, AAS, AAU, ABH, ACAD, AD, AK, AKPM, ALTA, AMNH, ARAN, ASU, B, BABY, BAS, BC, BCN, BEREA, BG, BM, BOON, BOUM, BR, BRI, C, CAN, CANB, CAS, CBM, CDBI, CHAS, CHR, CHRB, CHSC, CICY, CNS, COA, COFC, COI, COL, COLO, CONC, CORD, CP, CU, CVRD, DAO, DAV, DBG, DNA, E, EA, EKY, EMMA, ER, F, FAU, FCO, FLAS, FR, FRU, FTG, FULD, FURB, G, GAT, GB, GH, GLM, GMNHJ, GZU, HAL, HAST, HIB, HNT, HO, HSS, HU, HUJ, HYO, IAC, IBK, IBSC, ICEL, ICESI, ICN, INEGI, INM, INPA, IPA, ISKW, JBAG, JBGP, JCT, KYO, JYV, K, KMN, KOR, KPM, KSTC, KTU, KU, KUN, LAGU, LBG, LD, LE, LEB, LSU, LTB, LTR, MA, MACF, MAF, MAK, MB, MBK, MBM, MEL, MELU, MFU, MGC, MICH, MIN, MMMN, MNHM, MNHN, MO, MSB, MSC, MT, MTMG, MUB, NAC, NAS, NCSC, NCU, NE, NH, NHM, NHT, NMNL, NMR, NMSU, NSPM, NSW, NU, NY, O, OBI, OSA, OSC, P, PAMP, PE, PERTH, POM, QFA, QUE, REG, RELC, RNG, RSA, S, SALA, SAN, SANT, SAPS, SASK, SBBG, SCFS, SD, SDSU, SFV, SJSU, STU, SVG, TAI, TAIF, TALL, TAM, TAMU, TAN, TASH, TEF, TENN, TEX, TI, TKPM, TNS, TOYA, TRH, TROM, TRT, TRTE, UADY, UAM, UBC, UBT, UCR, UCS, UCSB, UCSC, UESC, UFG, UFMA, UFMT, UFS, UJAT, ULM, ULS, UME, UNB, UNM, UPNA, UPS, US, USCH, USF, USP, UT, UTEP, UWO, V, VAL, VIT, VT, W, WAG, WAT, WII, WIN, WOLL, WU, YAMA, Z, ZT, accessed via the BIEN dataset.

## Author contributions

A.R. and D.Z. conceived the ideas and designed the methodology; L.G. and P.W. helped refine the methodology; A.R., V.H., and K.S. prepared the data; A.R. analyzed the data; A.R. led the writing of the manuscript. A.R., V.H., K.S., L.G., T.K., P.W., D.C., J.S.C., and D.Z. contributed critically to the drafts and gave final approval for publication.

## Funding

## Competing interests

The authors declare no competing interests.
