## [Transparent Peer Review file · Nature Communications]

Climatic niche conservatism in non-native plants depends on introduction history and biogeographic context

Corresponding Author: Ms Anna Rönnfeldt

Version 0:

Reviewer comments:

Reviewer #1

(Remarks to the Author)
Rönnfeldt et al.
Nature Communications

This is an interesting manuscript on an important problem – quantifying the niche dynamics of introduced species, testing whether patterns are consistent with niche conservatism or niche switching and determining whether niche dynamics are contingent on traits or biogeography. Niche stability was relatively common and niche expansion was rare, unfilling seemed to be related to residence time. Traits were found to be less important influencers of niche dynamics than region, biogeographic attributes and residence time. The main finding is that inference of niche dynamics depends on region and residence time, with less unfilling for species with long residence time (which makes sense). However; the fundamental question of whether the regional context-dependency is due to real variation in biogeographic variables between regions or whether it is due to the idiosyncracies of the sample used (& small sample sizes for some regions) is still open in my opinion. Some commentary on this issue would be welcome.

The authors rightly point out that there are conflicting results on niche stability in introduced plants and this is a useful study in this field. It is not conclusive however.

The terms used “niche conservation” and “niche similarity” seem to be used interchangeably throughout. Please choose one term for clarity. However different results are presented for niche conservatism (45.5%) and niche stability (78.4%) and I was confused why these different percentages were used. Perhaps I’m missing some critical difference between how niche conservatism and niche stability are calculated?

Title: title is informative and results-based. As this study is anchored on a single region (Pacific Islands) this should be mentioned in the title.

Abstract:

L47-48 it is not clear here whether the study is restricted to the Pacific Islands or whether the PI are just one of eight broader regions, if so why call out this one region? There needs to be a clear statement in the abstract that the 316 species are a sample introduced to the PI and subsequently investigated across all other regions they have been introduced to.

Intro:

Generally a clear and well written introduction to the topic with hypotheses posed justified. However the justification for using the PI as a focal region is not entirely clear. For example:

L111-112: “a steep increase in invasive species per unit area” I’m not sure what this increase is. Do you mean that the number of invasive species increases per unit area? Or is this relationship steeper in the Pacific Islands compared to other regions?

L115: need to clarify here that the 316 spp are introduced to the PI and were matched with native or non-native status globally in the defined regions.

L117: clarify that you used an ordination approach to bioclimatic data based around occurrences in the different regions.

Results: for Fig 4 it would be helpful to integrate some of the expectations from table 1 into this figure to aid comparison with the expectations. For example you could colour the rows by whether you expect a positive (transparent red) or negative (transparent blue) effect.

L131: it is not clear that the sample of species were anchored on a sample of introduced plant species in the PI and subsequently compared across the other regions.

L137: it is not clear what the 30% figure relates to (the combination of tropical & temperate?), why highlight this category when tropical alone or temperate alone are also approx. 30%?

L138: not clear what you mean by an “even flow” can you be more precise?

L141: significant niche conservatism across all regions? Be specific here as it's not clear when you are referring only to the PI and when you are making a general statement across all regions.

L144-145: non-significance does not necessarily mean an “intermediate degree of niche changes”, surely it just means that any observed changes were not statistically supported? “an intermediate degree” is akin to an effect size rather than statistical significance.

L165 onwards, it would be useful to have the number of species here as well as the %.

L167: significance is narratively reported but no evidence is provided in text (test statistics, df, effect size, p-value...).

L191: while % variance explained is provided for traits and it is stated that biogeographic variables were more important the % variance explained was not provided.

Fig 4: the sample size relative to the number of variables is relatively low for some models (i.e. Europe, Sam, Atr all <80 for 10 explanatory variables).

Methods:

Given that the central data-set is anchored on plants introduced to the Pacific Islands, are there limitations to the study that stem from this initial sample? I.e. does the sample of plants include bias towards tropical species, species from a particular source region etc? The justification for this anchoring is not clear. Are these islands recipients of an average/non-average number of non-native plants compared to comparable regions?

L382: did the buffer include barriers to dispersal (e.g. ocean) which would be problematic for the assumption that these points represent biogeographically accessible areas? This might be particularly problematic for island occurrences.

(Remarks on code availability)

Reviewer #2

(Remarks to the Author)

Climatic niche conservatism in non-native plants depends on introduction history and biogeographic context

Niche conservatism is a key assumption in models predicting non-native species spread, but its generality is debated due to mixed empirical results. The manuscript by Rönnfeldt et al suggests that these contradictions arise from underexplored context dependency, particularly in species introduced to multiple regions. Using an ordination-based method, the authors have analyzed the climatic niche shifts in 1,566 introductions of 316 non-native plant species across eight regions, including the Pacific Islands.

Their findings show that regional context, biogeographic traits, and residence time help better explain niche changes than ecological traits. While their study showed that overall niche expansion was limited, the niche conservatism and niche unfilling (incomplete colonisation) varied by region. Species with smaller native ranges showed more expansion, and longer residence times reduced unfilling—implying that apparent niche shifts may be temporary and linked to dispersal limits. Their studies have highlighted the importance of regional context in understanding niche dynamics and improving models for risk assessment and management of non-native species.

The manuscript is well written, and the authors need to be applauded for the detailed study and analysis. However, the manuscript can be improved further if the authors attend to the comments and provide clarifications for some of the points listed below.

a) The rationale for selecting a 200 km terrestrial buffer around presence points to delineate biogeographically accessible areas is unclear. This distance appears quite large and may not accurately reflect ecological or dispersal realities for many plant species (especially in islands!). The authors should clarify the justification for this choice.

b) For widespread non-native species with large introduced ranges, such as *Lantana camara*, *Pontederia crassipes* or *Mikania micrantha* a focused discussion on how such species might have expanded (niche expansion) or shifted their niches (niche switch?)—along with the traits that may have facilitated this would strengthen the manuscript. This would also provide insight into the mechanisms behind successful invasions and/or range expansions of some of these invasive species.

c) While the authors consider plant traits, the treatment of ecological traits appears incomplete. In particular, dispersal mode (e.g., passive, wind, water, animal-mediated) is a critical trait influencing the potential for range shifts and colonization. This trait directly affects dispersal distance and should be incorporated into the trait analysis, or at the very least, discussed in terms of its implications for niche shifts and geographic expansion. Its omission limits the ecological interpretation of the observed patterns.

d) The manuscript substitutes the commonly used term “niche shift” with “niche switching,” but does not clearly explain the rationale or the specific magnitude or nature of change observed that warrants this terminological shift. Given that no species in the analysis exhibited significant niche switching, it would be useful for the authors to explain this conceptual choice more thoroughly. Moreover, a discussion on potential future niche switches—perhaps speculative, based on trait syndromes or ongoing range dynamics—would add depth. Are there species poised for niche switching under future climate change scenarios?

e) Figure S7, which illustrates the effects of traits on niche change metrics in non-native plants, presents valuable insights and deserves inclusion in the main manuscript. Its relevance to the central arguments supports its promotion to a main figure. Figure S8 could continue to be retained as a supplementary figure.

f) What has been the niche dynamics in congeneric species, which share several traits? Since the data set includes several congeneric species, it would be worth exploring how some species have range expansions and others in the same genus have their niche conserved.

g) Could the authors also provide the list of the 20 species which consistently have their climatic niche conserved? Similarly, the list of 47 which showed no niche conservatism and species which showed more than 50% niche expansion (as another supplementary table?)

h) The authors found that the niche unfilling tended to decrease and niche stability increase for long lived species – this needs some clarity and mention in the discussion. Similarly, it is not clear the association between plant height and niche expansion?.

(Remarks on code availability)

Version 1:

Reviewer comments:

Reviewer #1

(Remarks to the Author)

The authors have adequately addressed my concerns and suggestions. The manuscript is clear and compelling.

(Remarks on code availability)

Reviewer #2

(Remarks to the Author)

The authors have carefully revised the manuscript in response to the suggestions provided. I appreciate their effort in addressing each comment thoroughly, offering clear explanations, and incorporating the necessary changes into the revised version. The revisions have strengthened the manuscript, and I am satisfied with the responses provided.

However, regarding Figure 4, I believe it contains important information that adds value to the overall narrative. Therefore, I suggest that it should be included in the main text rather than being placed as supplementary material.

(Remarks on code availability)

Dear reviewers,

Thank you very much for taking the time to review our manuscript. We sincerely appreciate the thoughtful comments and suggestions, which have helped us to improve the clarity and robustness of the work.

In the revised version, we have carefully revised the text and figures for improved clarity and presentation. In addition, we conducted new sensitivity analyses to explicitly test the effects of key modelling decisions.

Please find our detailed point-by-point responses below.

Anna Rönnfeldt

Reviewer #1 (Remarks to the Author):

Rönnfeldt et al.
Nature Communications

This is an interesting manuscript on an important problem – quantifying the niche dynamics of introduced species, testing whether patterns are consistent with niche conservatism or niche switching and determining whether niche dynamics are contingent on traits or biogeography. Niche stability was relatively common and niche expansion was rare, unfilling seemed to be related to residence time. Traits were found to be less important influencers of niche dynamics than region, biogeographic attributes and residence time. The main finding is that inference of niche dynamics depends on region and residence time, with less unfilling for species with long residence time (which makes sense). However; the fundamental question of whether the regional context-dependency is due to real variation in biogeographic variables between regions or whether it is due to the idiosyncracies of the sample used (& small sample sizes for some regions) is still open in my opinion. Some commentary on this issue would be welcome.

Response: Thank you for this overall positive assessment. We provide a more detailed account of our changes below. Note that the line numbers we provide refer to the revised manuscript version with the track changes. We also submitted a clean version without the track changes, should you prefer it for reading.

The authors rightly point out that there are conflicting results on niche stability in introduced plants and this is a useful study in this field. It is not conclusive however.

The terms used “niche conservation” and “niche similarity” seem to be used interchangeably throughout. Please choose one term for clarity. However different results are presented for niche conservatism (45.5%) and niche stability (78.4%) and I was confused why these different percentages

were used. Perhaps I'm missing some critical difference between how niche conservatism and niche stability are calculated?

Response: We carefully revised the text to clarify these questions. Niche conservatism is assessed using a similarity test, and we now more clearly define this in the text and exclusively use the word similarity when referring to the actual null model test, and we otherwise stick to niche conservatism. Niche conservatism and niche stability are two different aspects, and we revised the text to more clearly describe the difference: We assume niche conservatism for a species if the similarity test, a null model test, indicates that the native and non-native niches overlap more strongly than would be expected by chance. Contrary to that, niche stability is one of the niche change metrics describing the niche dynamics between native and non-native niches; these metrics are derived by comparing native and non-native niche overlap and report the proportions of niche space that are overlapping or not. More specifically, niche stability describes the proportion of the niche that is occupied both in the native and the non-native range. Our niche comparisons indicated that the median niche stability across niche pairs is 78.4%, meaning that, on average, 78.4% of the niche space was consistent between native and non-native range. Additionally, the similarity tests inform about significance and revealed that 45.5% of the analysed niche pairs (n=1566) exhibited niche conservatism, meaning a significantly higher niche stability than expected by chance. These definitions follow previous publications on this issue, e.g., Guisan et al. 2014 (<https://doi.org/10.1016/j.tree.2014.02.009>) and Broennimann et al. 2007 (<https://doi.org/10.1111/j.1461-0248.2007.01060.x>).

Title: title is informative and results-based. As this study is anchored on a single region (Pacific Islands) this should be mentioned in the title.

Response: We appreciate this suggestion, but we would prefer to keep the more general title. Although the initial species list considers species that have been introduced to the Pacific, all of these species are also introduced in other parts of the world, and this multi-region comparison is one of the novel aspects of our study. As our results apply to multiple geographical regions, the conclusions go well beyond the Pacific region and, in our view, justify the more general title.

Abstract:

L47-48 it is not clear here whether the study is restricted to the Pacific Islands or whether the PI are just one of eight broader regions, if so why call out this one region? There needs to be a clear statement in the abstract that the 316 species are a sample introduced to the PI and subsequently investigated across all other regions they have been introduced to.

Response: Thank you for pointing this out. We agree that mentioning the Pacific islands in the abstract could be confusing. We explicitly include the Pacific region as one of the regions in our study because we assume that niche dynamics differ between dispersal-limited archipelagos and more continental regions

with comparably lower dispersal limitations. We have now changed the abstract to explicitly mention continents vs. archipelagos, rather than the Pacific region. [L50]

Intro:

Generally a clear and well written introduction to the topic with hypotheses posed justified. However the justification for using the PI as a focal region is not entirely clear. For example:

L111-112: “a steep increase in invasive species per unit area” I’m not sure what this increase is. Do you mean that the number of invasive species increases per unit area? Or is this relationship steeper in the Pacific Islands compared to other regions?

Response: The statement refers to both the number of invasive species increasing per unit area and that increase being steeper compared to other regions. We revised the sentence to clarify further. [L116-118]

L115: need to clarify here that the 316 spp are introduced to the PI and were matched with native or non-native status globally in the defined regions.

Response: Thank you. We revised this paragraph to improve clarity. [L122-125]

L117: clarify that you used an ordination approach to bioclimatic data based around occurrences in the different regions.

Response: We revised the text accordingly. [L126-127]

Results:

for Fig 4 it would be helpful to integrate some of the expectations from table 1 into this figure to aid comparison with the expectations. For example, you could colour the rows by whether you expect a positive (transparent red) or negative (transparent blue) effect.

Response: Thank you for the suggestion to add a visual aid regarding the underlying hypotheses. We agree that it will help with the interpretation of our results. We tried colouring the underlying rows as

suggested, but found this too dense. Rather, we have now added an additional column to each panel with line graphs indicating the hypothesised effect direction in following Table 1. [L269]

L131: it is not clear that the sample of species were anchored on a sample of introduced plant species in the PI and subsequently compared across the other regions.

Response: We clarified this accordingly. [L148-L150]

L137: it is not clear what the 30% figure relates to (the combination of tropical & temperate?), why highlight this category when tropical alone or temperate alone are also approx. 30%?

Response: We revised this part and removed the 30% figure to improve clarity. [L157-160]

L138: not clear what you mean by an “even flow” can you be more precise?

Response: We revised this part to improve clarity. [L157-160]

L141: significant niche conservatism across all regions? Be specific here as it’s not clear when you are referring only to the PI and when you are making a general statement across all regions.

Response: We considerably revised this part to improve the clarity. [L167 and following]

L144-145: non-significance does not necessarily mean an “intermediate degree of niche changes”, surely it just means that any observed changes were not statistically supported? “an intermediate degree” is akin to an effect size rather than statistical significance.

Response: We revised the text accordingly by removing the section with the “intermediate degree of niche changes”. [L173]

L165 onwards, it would be useful to have the number of species here as well as the %.

Response: The percentage values in this section do not refer to the percentage of species, but instead the median values of the niche metrics. We have now revised this part to improve clarity. [L198-203]

L167: significance is narratively reported but no evidence is provided in text (test statistics, df, effect size, p-value...).

Response: We have now added the missing test details in the text. [L207]

L191: while % variance explained is provided for traits and it is stated that biogeographic variables were more important the % variance explained was not provided.

Response: We have now added the percentage values for the median variable importance of the ecological traits for the three niche metrics. We decided against listing the % values for the other two categories as well, to improve the readability of the section. The magnitude of these values can be deduced from the values listed for the ecological traits, which can also be found in Fig. 4. [L233-L235]

Fig 4: the sample size relative to the number of variables is relatively low for some models (i.e. Europe, Sam, Atr all <80 for 10 explanatory variables).

Response: Although we agree that sample size is at the lower end of the recommended sample size for regressions, we are confident that our analyses are robust in this respect. First, most statistical textbooks recommend 5-10 data points per predictor variable, which we achieve for seven out of eight regions. Second, we fitted univariate models (allowing the use of more data points) as an additional test, which are consistent in terms of effect direction with those of the multiple regression models (Fig. S12). We are thus confident that our conclusions are robust against sample size issues.

Methods:

Given that the central data-set is anchored on plants introduced to the Pacific Islands, are there limitations to the study that stem from this initial sample? I.e. does the sample of plants include bias towards tropical species, species from a particular source region etc? The justification for this anchoring is not clear. Are these islands recipients of an average/non-average number of non-native plants compared to comparable regions?

Response: As shown in Fig. S2, the selected species originate from different climate zones, with equal proportions from temperate and tropical zones, and these species also spread equally across regions. We are confident that focusing the original species selection on the Pacific islands has reduced bias in terms of the species selection, as all of the studied species have also been introduced elsewhere. This ensures that the Pacific region, as a large group of archipelagos, is well represented in the data. Basing our species selection on other regions might have resulted in an underrepresentation of dispersal-limited archipelagos. Thus, our final species selection included both temperate and tropical species, from various plant families and biogeographic origins (their native ranges often covering multiple countries or even continents) [L468-470].

L382; did the buffer include barriers to dispersal (e.g. ocean), which would be problematic for the assumption that these points represent biogeographically accessible areas? This might be particularly problematic for island occurrences.

Response: Thank you for raising this important point. We agree that the selection of a spatial buffer for generating the background data is, to some degree, arbitrary, yet it is a common approach in niche modelling, and we also consider it the most pragmatic approach here. We chose 200 km as a spatial buffer to adequately account for the biogeographic (historic) dispersal ability in the native range and the potentially human-mediated dispersal in the non-native ranges. We have not considered barriers such as larger water bodies, deserts, mountain ranges or anthropogenic structures. We agree that the ocean could constitute a barrier to the dispersal of many species, but certainly not to all species (e.g., *Cocos nucifera*). Consequently, as a sensitivity analysis, we tested the effects of using a smaller spatial buffer of 50 km for background data. The smaller buffer would, to some extent, account for stronger dispersal limitations. We randomly chose 15% of the studied species ($n = 49$) and repeated all analyses using the 50 km buffer (see Figs. S5-S8 for results). These analyses largely corroborated our previous results or even increased previously reported differences between regions: Pacific Islands are the region with the lowest regional percentage of species with significant niche conservatism (Fig. S5), and with higher niche unfilling compared to other regions (Fig. S7). We thus conclude that our results are robust to the choice of the buffer size. We thus chose to continue using the 200 km buffer in the main analyses and manuscript, as we consider the 50 km to be unrealistically low for the native regions and also species introduced to mainland regions where dispersal limitation is low. We now mention results of the sensitivity analysis in the methods section in the revised version of the manuscript, and added new figures S5-S8 in the appendix. [L480-484]

Reviewer #2 (Remarks to the Author):

Climatic niche conservatism in non-native plants depends on introduction history and biogeographic context

Niche conservatism is a key assumption in models predicting non-native species spread, but its generality is debated due to mixed empirical results. The manuscript by Rönnfeldt et al suggests that these contradictions arise from underexplored context dependency, particularly in species introduced to multiple regions. Using an ordination-based method, the authors have analyzed the climatic niche shifts in 1,566 introductions of 316 non-native plant species across eight regions, including the Pacific Islands.

Their findings show that regional context, biogeographic traits, and residence time help better explain niche changes than ecological traits. While their study showed that overall niche expansion was limited, the niche conservatism and niche unfilling (incomplete colonisation) varied by region. Species with smaller native ranges showed more expansion, and longer residence times reduced unfilling—implying that apparent niche shifts may be temporary and linked to dispersal limits. Their studies have

highlighted the importance of regional context in understanding niche dynamics and improving models for risk assessment and management of non-native species.

The manuscript is well written, and the authors need to be applauded for the detailed study and analysis. However, the manuscript can be improved further if the authors attend to the comments and provide clarifications for some of the points listed below.

Response: We thank the reviewer for this positive assessment. We provide a more detailed account of our changes below. Note that the line numbers we provide refer to the revised manuscript version with the track changes. We also submitted a clean version without the track changes, should you prefer it for reading.

a) The rationale for selecting a 200 km terrestrial buffer around presence points to delineate biogeographically accessible areas is unclear. This distance appears quite large and may not accurately reflect ecological or dispersal realities for many plant species (especially in islands!). The authors should clarify the justification for this choice.

Response: Thank you for raising this important point. We agree that the selection of a spatial buffer for generating the background data is, to some degree, arbitrary, yet it is a common approach in niche modelling, and we also consider it the most pragmatic approach here. We chose 200 km as a spatial buffer to adequately account for the biogeographic (historic) dispersal ability in the native range and the potentially human-mediated dispersal in the non-native ranges. We have not considered barriers such as larger water bodies, deserts, mountain ranges or anthropogenic structures. We agree that the ocean could constitute a barrier to the dispersal of many species, but certainly not to all species (e.g., *Cocos nucifera*). Consequently, as a sensitivity analysis, we tested the effects of using a smaller spatial buffer of 50 km for background data. The smaller buffer would, to some extent, account for stronger dispersal limitations. We randomly chose 15% of the studied species ($n = 49$) and repeated all analyses using the 50 km buffer (see Figs. S5-S8 for results). These analyses largely corroborated our previous results or even increased previously reported differences between regions: Pacific Islands are the region with the lowest regional percentage of species with significant niche conservatism (Fig. S5), and with higher niche unfilling compared to other regions (Fig. S7). We thus conclude that our results are robust to the choice of the buffer size. We thus chose to continue using the 200 km buffer in the main analyses and manuscript, as we consider the 50 km to be unrealistically low for the native regions and also species introduced to mainland regions where dispersal limitation is low. We now mention results of the sensitivity analysis in the methods section in the revised version of the manuscript, and added new figures S5-S8 in the appendix. [L480-484]

b) For widespread non-native species with large introduced ranges, such as *Lantana camara*, *Pontederia crassipes* or *Mikania micrantha* a focused discussion on how such species might have expanded (niche expansion) or shifted their niches (niche switch?) —along with the traits that may have facilitated this

would strengthen the manuscript. This would also provide insight into the mechanisms behind successful invasions and/or range expansions of some of these invasive species.

Response: Thank you for this suggestion. Of the species mentioned, only *Lantana camara* was included in our study. While *L. camara* is indeed widespread across its non-native range, this did not translate into significant niche expansion in our analysis. Instead, this species exhibited strong niche conservatism across most regional introductions (see supplementary data table). We recognise that the potential of establishing a large non-native range offers an interesting angle and we appreciate the suggestion. However, we would like to clarify that our study did not assess range size in the non-native range, and thus does not directly address invasion success or spread. Instead, we focused on climatic niche dynamics, particularly the extent of niche stability, unfilling, and expansion across regions. Interestingly, we found that species with smaller native ranges exhibited greater niche expansion, potentially reflecting release from biotic constraints in the introduced range. While we agree that a trait-based exploration of mechanisms behind invasion success would be valuable, this is well beyond the scope of our current study. We believe such questions may be better addressed using targeted species- or community-level analyses, as we now point out in our revised discussion. [L430-435]

c) While the authors consider plant traits, the treatment of ecological traits appears incomplete. In particular, dispersal mode (e.g., passive, wind, water, animal-mediated) is a critical trait influencing the potential for range shifts and colonization. This trait directly affects dispersal distance and should be incorporated into the trait analysis, or at the very least, discussed in terms of its implications for niche shifts and geographic expansion. Its omission limits the ecological interpretation of the observed patterns.

Response: We agree that more detailed trait information on dispersal would enhance the ecological interpretation of our results. Unfortunately, such data were limited or inconsistent for many of the species in our dataset. Moreover, available dispersal mode classifications (e.g., anemochory vs. hydrochory) do not always translate into meaningful differences in dispersal distance, which complicates their interpretation in a comparative framework. Meanwhile, plant height and the other proxy traits we selected correlate well with dispersal distance (Bullock et al. 2017, <https://doi.org/10.1111/1365-2745.12666>; Thomson et al. 2017, <https://doi.org/10.1111/nph.14735>). Early and Sax (2014, <https://doi.org/10.1111/geb.12208>), using a method by Vittoz and Engler (2007), characterised dispersal ability based on traits such as plant height, dispersal mode and vectors. We were unable to replicate this approach due to incomplete trait data for our focal species. Notably, Early and Sax (2014) found no relationship between dispersal ability and niche dynamics, which is consistent with our findings. In response to this comment, we have now clarified in the Methods section why dispersal mode was not included in our analyses [L496-499]. We also expanded the discussion to acknowledge that future studies, particularly those focused on finer spatial scales, could benefit from incorporating additional ecological traits related to both dispersal and competitive ability, where data availability and contextual detail allow for more meaningful interpretation. [L426-439]

d) The manuscript substitutes the commonly used term “niche shift” with “niche switching,” but does not clearly explain the rationale or the specific magnitude or nature of change observed that warrants

this terminological shift. Given that no species in the analysis exhibited significant niche switching, it would be useful for the authors to explain this conceptual choice more thoroughly. Moreover, a discussion on potential future niche switches—perhaps speculative, based on trait syndromes or ongoing range dynamics—would add depth. Are there species poised for niche switching under future climate change scenarios?

Response: We apologize for any confusion. As we now make more explicit in the revised text, niche switching refers to significantly lower niche overlap than expected by chance [L567-570; L570-573 now deleted for clarity]. In contrast, the term “niche shift” does not imply any statistical test, but simply refers to differences in the occupied niche space between native and non-native regions. As we explicitly analyse the niche change metrics niche stability, niche unfilling and niche expansion, we avoided using the ambiguous term “niche shift”. Our results indicated no significant niche switching. We expanded our discussion to address the potential influence of climate change. [L398-402]

e) Figure S7, which illustrates the effects of traits on niche change metrics in non-native plants, presents valuable insights and deserves inclusion in the main manuscript. Its relevance to the central arguments supports its promotion to a main figure. Figure S8 could continue to be retained as a supplementary figure.

Response: Thank you for this suggestion. We have tried different ways of incorporating the additional information into the figure in the main text. Merging Fig. 4 and S11 (previously S7) would result in a considerably larger figure, and we are not convinced that the additional information in panels D-F warrants the required space. We would thus like to leave the decision of potentially replacing Fig. 4 with the 6-panel figure (see below) to the discretion of the editor. Please note that panels D-F only show the effect size and direction, but not variable importance, as the low importance of some variables in the full model would make the circles hardly discernible.

Figure 4. Trait effects on climatic niche changes in non-native plants. For each non-native region, we fitted regression models with AIC-based stepwise selection for (A) niche unfilling, (B) niche stability and (C) niche expansion. Panels D-F show the full regional models. The first column (H1) in each panel shows our hypotheses about the expected relationship (see Tab. 1) between the respective traits and the niche metrics where applicable. All other columns represent the final model for each region, with the total explained variance (R^2) shown below the columns. Circle sizes indicate the variable importance (%), not shown for the full models in D-F) of single traits within the multiple regression models, and the effect size shows whether the respective niche change metric increases (red) or decreases (blue) as the trait values increase. Traits

labelled with an asterisk refer to biogeographic traits estimated for the native range or native niche of the species. Niche centroids refer to the relative position along climatic gradients 1 (from warm to cold) and 2 (from wet to dry). The sample size varied between regions: Africa (Afr, n = 124), temperate Asia (Ate, n = 95), tropical Asia (Atr, n = 78), Australasia (Aus, n = 124), Europe (Eur, n = 56), North America (Nam, n = 110), Pacific Islands (Pac, n = 143), South America (Sam, n = 41).

f) What has been the niche dynamics in congeneric species, which share several traits? Since the data set includes several congeneric species, it would be worth exploring how some species have range expansions and others in the same genus have their niche conserved.

Response: Thank you for pointing this out. We have used phylogenetic regression to account for phylogenetic effects that are not explained by the considered traits. We have now added information on the strength of the phylogenetic signal in each regional model to the supplementary materials (Table S2). While for many regions, the phylogenetic signal approached zero, we also saw high variability across niche change metrics and regions. This underscores the high context dependency of niche conservatism and shows that for some regions, the considered traits are not sufficient to capture all variability across congeneric species. We have now expanded the discussion to address this. [L428-439]

g) Could the authors also provide the list of the 20 species which consistently have their climatic niche conserved? Similarly, the list of 47 which showed no niche conservatism and species which showed more than 50% niche expansion (as another supplementary table?)

Response: The names of these species are now listed in a supplementary data file.

h) The authors found that the niche unfilling tended to decrease and niche stability increase for long lived species – this needs some clarity and mention in the discussion. Similarly, it is not clear the association between plant height and niche expansion?.

Response: Life cycle and plant height are indeed important predictors in some regions, but not in all. We have now expanded the discussion and highlighted that these are regional exceptions and that no general patterns emerged for them across all regions [L414-417]. We further discuss that including data on additional factors, such as horticultural use or anthropogenic disturbances, could contribute to our understanding of the unexpected regional importance of these factors. [L435-L439]

Dear reviewers,

Thank you again, on behalf of the entire author team, for your feedback and for providing your time to review our manuscript!

Below, we provide our point-by-point response to the last open remark by reviewer #2.

Anna Rönnfeldt

Reviewer #1 (Remarks to the Author):

The authors have adequately addressed my concerns and suggestions. The manuscript is clear and compelling.

Reviewer #2 (Remarks to the Author):

The authors have carefully revised the manuscript in response to the suggestions provided. I appreciate their effort in addressing each comment thoroughly, offering clear explanations, and incorporating the necessary changes into the revised version. The revisions have strengthened the manuscript, and I am satisfied with the responses provided.

However, regarding Figure 4, I believe it contains important information that adds value to the overall narrative. Therefore, I suggest that it should be included in the main text rather than being placed as supplementary material.

Response: Following your recommendation, we have now added the updated version of Figure 4 (now including the full models for niche unfilling, stability, and expansion) to the main text.